# The Consequences of Child Abuse

**DOI:** 10.3390/healthcare11111650

**Published:** 2023-06-05

**Authors:** Ami Rokach, Shauna Clayton

**Affiliations:** 1Psychology Department., Faculty of Health, York University, Toronto, ON M3J 1P3, Canada; 2Department of Kinesiology and Health Science, Faculty of Health, York University, Toronto, ON M3J 1P3, Canada

**Keywords:** children, maltreatment, abuse, consequences

## Abstract

This review provides an overview of the consequences of early adverse experiences across various domains of life. Drawing on the Adverse Childhood Experiences (ACEs) conceptual framework, we discuss the ACE pyramid and the varying degrees of consequences that ACE exposure may elicit. Using online search engines such as Google Scholar, the authors sifted through empirical research to locate relevant articles and research to help prepare this review. This article sheds light on the implications of ACEs for health, socio-emotional and psychosocial well-being, relationships, personality, and cognitive functioning.

## 1. Introduction

Child maltreatment, including physical, sexual, and emotional abuse and neglect, is a global phenomenon. Adverse Childhood Experiences (ACEs) is a conceptual framework based on decades of research that highlights how early experiences such as maltreatment impact development and incite negative health outcomes across the lifespan. Both academic and public sources indicate that ACEs can be found in all families and at all socioeconomic levels, with consequences that are far-reaching [1]. The present review aims to provide a comprehensive overview of the consequences of child abuse. To include every consequence of child maltreatment would go beyond the scope of this paper; however, we provide insight into some main risk factors and associated consequences.

Research suggests that the causal pathway to adverse life outcomes or, in severe instances, early death can be traced back to early instances of childhood trauma [2]. In their landmark study on this very topic, Felitti et al. [2] discovered how real the link between early negative experiences and poor health can be for individuals who have experienced various forms of maltreatment. Hence, to conceptualize this, a framework was invented that clinicians and researchers widely reference, referred to as Adverse Childhood Experiences (ACEs). The framework provides insight into the associated risks of trauma and health and is embedded within an eight-tier pyramid used to depict potential consequences [3]. The pyramid includes, from the base up, eight stages through which ACEs and their effects might progress: (8) generational embodiment/historical trauma; (7) social conditions/context; (6) child abuse experiences; (5) neurodevelopment disruptions; (4) impairment of the social, emotional and cognitive dimensions; (3) practicing health-wise risky behaviors; (2) the connection between disease, disability, and social problems; (1) early death. Beginning with the bottom stage, the first three stages address various forms of exposure, which, if left unmitigated, can segue into subsequent stages. ACE exposure may result in toxic stress if the exposure is continuous, which can result in neurodevelopmental deficits and impairment of social, emotional, and cognitive functioning [4]. Disruptions to bio-psychosocial functioning may result in behavioral consequences, such as health risk behaviors that may negatively impact life outcomes, including physical and mental health, and perhaps result in early death. Thus, it was suggested that the onset of disease or illness subsequent to health risk behaviors might have emotional connections to earlier unresolved trauma. The residual impact of such occurrences and toxic stress, if continuous, may lead to maladaptive coping mechanisms [5,6]. For instance, an inability to cope with emotional pain stemming from childhood abuse may result in increased alcohol consumption, which research suggests is linked to early death [7].

Trauma can impact individuals, families, and even entire communities. Trauma can stem from various forms of abuse and neglect that an individual may experience but is sometimes also a consequence of generational legacy: the collective experiences that result in intergenerational trauma [5]. Though the catalysts of trauma can vary, research suggests that we can find increased occurrences of physical health issues such as obesity, cardiovascular disease, and stroke in marginalized communities [8]. Although these outcomes have genetic components, environmental influences seem to also have an impact on disease pathology. Take, for example, inaccessible healthy food options, which, owing to economic instability, may lead to a poor diet. Moreover, in such communities, individuals have less access to information and awareness of topics that may benefit them and their way of life. In another example, when violence is part of a community’s way of life, it is found to be linked to depressive and anxiety disorders and PTSD [9]. Without a support system to facilitate resilience among individuals, it may be difficult for families to overcome community-related obstacles, and they subsequently pay the price for it in their homes. The effects of trauma can become so profound that it may create a coercive cycle of repeated exposure to trauma-inducing experiences, the same or new ones, often because the impact goes unresolved [10]. Moreover, examples of intergenerational trauma are those that involve the experience of childhood abuse or domestic violence that may be perpetuated across generations [11]. Moreover, individuals who have witnessed substance use or domestic violence in their childhood homes may engage in the same behavior when they are older. These examples of trauma stemming from social conditions and local context are highlighted in the ACE pyramid as potential catalysts to a perpetuation of further challenges across the lifespan. This is important to recognize as social conditions or the social environment address the systemic barriers that create disadvantages because of the community [12]. For example, disparities in race, ethnicity, or religion subject individuals to discrimination and marginalization. Therefore, intervention in communities where individuals are at an increased risk for exposure to poverty or community violence is imperative to mitigate the negative effects on children [13]. It is now well known that children require safe and nurturing environments to develop resilience. When social conditions are unfavorable for optimal development, children suffer consequences throughout childhood and adulthood [14]. Finally, to this point, beyond the community level larger-scale geographical conditions, such as cultural values and parenting norms in various communities, may also influence the occurrence of ACEs [15,16]. These components of the ACE conceptual framework underscore the imperativeness of incorporating a lens that recognizes the collective environment as a potential factor in individual experiences.

The geographical, historical, generational, and social contexts in which the child is raised may be precursors to ACEs and the child’s developmental trajectory, as affected by historical, generational, social, and local factors. Under conditions of maltreatment, the child has an increased risk for neurodevelopmental disruptions and impairments to social, emotional, and cognitive functioning [17]. For instance, hyperactivation of the hypothalamic–pituitary axis and prolonged secretion of stress-related hormones such as cortisol may be precipitated by repeated exposure to trauma, resulting in toxic stress [6]. The neuroendocrine effects of ACE exposure may negatively affect the child’s ability to regulate emotions and behavior [6]. It was found that irregular neural structures that would otherwise facilitate healthy brain development and response systems can negatively affect a child’s social, emotional, and cognitive development. Such children may display problematic behaviors at home and school, decreasing their chance of developing healthy relationships [18]. For example, hypervigilance and an inability to regulate emotions may increase the risk that they will display peer-victimizing behavior such as aggression and suffer from memory and learning difficulties [18,19].

### Methodology

A comprehensive review was conducted through Google Scholar that accounted for eight possible keyword search queries, namely, child abuse; adverse childhood experiences; trauma; causes; consequences; implications; mental health; physical health; intimate partner violence. The guidelines followed for the search criteria ensured that the first set of keywords (i.e., child abuse) be in the title, while the other keywords could be anywhere in the text. By including duplicate articles that may have appeared in multiple search queries, the set of keywords produced 198 total articles in Google Scholar. Of these, 119 studies conducted globally and on various forms of child maltreatment were identified as appropriate for this topic and were referenced.

## 2. ACEs and Psychosocial Well-Being

Beutel et al. [20] contended that resilience “can be defined as an outcome in the face of adversity or as a process mediating the response to stress or trauma... Resilience factors are empirically derived variables that statistically predict a resilient outcome. Thus, they link two elements, the exposure to risk or hardship and a positive outcome within or higher than the expected range… The most prominent factor is an individual’s ability to respond positively to physiological, psychological, or social-environmental challenges. Resilient individuals have been shown to use effective, active problem-solving patterns… and adaptive appraisal styles in terms of coping mechanisms” (p. 2). The researchers investigated the buffering effect of resilient coping styles on mental health in people who were abused as children. Their results indicated that resilient people reported fewer depression, anxiety, and somatic symptoms, which frequently follow childhood maltreatment, than those experienced by vulnerable participants.

Resilience is important for children in general, but particularly for those experiencing medical problems, psychological difficulties, divorce, or loss of a parent [21,22]. Resilience is a process that may affect children’s interactions with adverse life conditions reducing their harmfulness. However, they are expected to have some distress symptoms, mainly if the adverse experiences were severe or long-lasting [23]. Children who are not resilient often lack the basic support, protection, and respect they need, whereas children with the required external support often succeed in continuing forward [24,25,26].

Without resilience, child abuse or neglect can significantly affect well-being and quality of life and result in health risk behaviors [24]. Well-being was defined by the World Health Organization (WHO) as “a positive state experienced by individuals and societies. Similar to health, it is a resource for daily life and is determined by social, economic, and environmental conditions” [27], para.1. Thus, psychosocial well-being can be viewed as a dimension of well-being that incorporates psychological well-being along with social and collective well-being [28]. Psychosocial deficits associated with ACEs contribute to disruptions in cognition, memory, and attention processing and increase the risk for psychiatric disorders [29,30], anxiety, and depressive disorders [31]. Having suffered adversity in childhood appears to decrease one’s ability to manage stressors and acquire the psychosocial resources needed to overcome challenges [32]. ACEs have also been found to increase the risk of re-victimization and negatively affect well-being [32]. It was shown that those who suffer ACEs are at increased risk for social isolation and loneliness and may succumb to early death [33,34,35]

## 3. Sexual Intimacy Difficulties

The trauma accompanying ACEs may profoundly impact an individual’s ability to trust others, and because of their inability to change parental abuse, they grow up feeling powerless and mistrustful [36]. This is important to consider when addressing issues related to child abuse as research has indicated that showing affection and engaging in sexual intimacy are associated with overall couple satisfaction in adulthood [37]. Sexual intimacy allows for closeness and vulnerability based on trusting another person. This ability depends on factors including self-esteem, sense of self-worth, and security [38]. Trust may promote self-disclosure, which intensifies intimacy [39]. Self-disclosure usually signifies a close bond, based on trust, between the individuals, and it facilitates an awareness that the partner cares about and understands the other’s thoughts and feelings [40]. Research indicates that intimacy is enhanced if relationship responsiveness is regularly reciprocated [41]. The intimacy process model emphasizes the importance of maintaining this interaction to improve intimacy over time. Otherwise, intimacy may decline and negatively affect the association and its longevity [42].

Openly engaging in a dialogue regarding thoughts and feelings can enhance sexual intimacy and relationship quality [43]. However, open discussion regarding emotions may be significantly hampered in those individuals who have difficulty trusting, getting emotionally close, and communicating openly about feelings. Research also suggests that child abuse can create confusion around what is safe or healthy behavior in relationships. Research regarding violent childhood experiences found that children who underwent physical or sexual abuse were at an increased risk of perpetrating violence toward their intimate partners [44]. Child sexual abuse (CSA) was also found to significantly interfere with attachment processes and the subsequent quality of relationships [45]. As CSA victims develop further and become adults, they may experience difficulty trusting and allowing emotional vulnerability with their partners. CSA was also positively associated with developing psychopathologies, such as substance use, mood disorders, anxiety, and PTSD [46,47]

## 4. Personality Disorders and Their Association with ACE

ACEs affect personality development and psychopathological outcomes [48]. For instance, a Chinese study found a significant association between adverse experiences in childhood and susceptibility to psychiatric disorders and various personality deficits [49]. Similarly, a relationship between ACEs and neuroticism increased openness, and of the big five personality traits, low extraversion was found [50,51]. Personality has been defined as “the dynamic integration of the totality of a person’s subjective experience and behavior patterns” [52]. It was established that the interaction between nature and nurture affects personality development [52]. ACEs appear to interact with temperamental factors that impact personality development [53]. ACEs may undermine developmental processes and psychological adjustment that will last into adulthood. For example, a study found that a relationship between parental antipathy and negative loving can increase the pathology of antisocial personality disorder [54]. Over time, personality deficits become dysfunctional and contribute to maladaptive behavioral and emotional output. Personality issues that go unattended tend to become problematic and may impact regulatory and self-functioning processes. The 5th edition of the Diagnostic Statistical Manual (DSM-5) defines a personality disorder as: “enduring a pattern of inner experience and behavior that deviates markedly from the norms and expectations of the individual’s culture, is pervasive and inflexible, has an onset in adolescence or early adulthood, is stable over time, and leads to distress or impairment” [55]. Research further suggests that the neurodevelopmental implications of ACEs may contribute to the etiology of psychopathological outcomes among victims of child abuse [56]. Thus, we can see a link between toxic stress and the development of personality disorders. Moreover, an increase in comorbidity with mood disorders, such as major depressive disorder, was observed among those subjected to ACEs. Additionally, there is a positive correlation between the number of ACEs the child was subjected to and the development of avoidant, obsessive, schizotypal, and borderline personality disorders [57,58]. Specific types of ACEs may precede specific psychopathological outcomes. For example, borderline personality disorder (BPD) seems highly associated with childhood sexual trauma, especially among women [6,59]. The DSM-5 states that BPD involves impulsive and self-destructive behavior, relational instability, and identity confusion [55].

In another example, histrionic personality disorder is another common personality disorder that may be associated with childhood trauma, including abuse and neglect [42]. Those with this personality disorder engage in attention-seeking behaviors and express exaggerated emotionality [60]. Moreover, adults who may have suffered maltreatment as children may also present with avoidant personality disorder (APD), characterized by an intense fear of inadequacy and concern that others impose malicious intent [55,61]. These individuals often avoid situations that might expose them to the very circumstances they fear. They desire intimate relationships, but their avoidance of others may hamper their attempts to create such relationships [55]. Finally, having suffered ACEs in childhood is also associated with antisocial personality disorder (ASPD) in adulthood [62]. These individuals tend to be deceitful, lack remorse, and cannot conform to social norms [63]. Furthermore, these individuals tend to commit theft or violent crimes [63]. Early sexual abuse, physical abuse, neglect, and separation from parents are closely associated with ASPD [63].

## 5. Psychiatric Consequences of Child Abuse

Research has indicated that a significant increase in major psychiatric disorders, including major depression, bipolar disorder, post-traumatic stress disorder (PTSD), and alcohol and drug abuse, is related to ACEs. Nemeroff [64], having reviewed many studies, stated that sexual, physical, and emotional abuse clearly and significantly contribute to adulthood mood and anxiety disorders and certain other medical disorders. Scott et al. through their research in New Zealand, found that ACEs were associated with increased PTSD, mood disorders, and substance use disorders [65]. These findings were confirmed by Putnam and colleagues as they found that multiple ACEs resulted in complex adult psychopathology [66].

Exploring the consequences of a single type of ACE on adult psychopathology, Chen et al. conducted a meta-analysis involving 37 studies [67]. They found a significant positive correlation between sexual abuse in both men and women and depression, eating disorders, PTSD, sleep disorders, and suicide attempts. Moreover, even bullying, which is basically the verbal aggression of parents toward their children, negatively influences children’s mental health, much like the effect of observing domestic violence and non-familial sexual abuse [22]. Several large-scale studies, which took place in the U.S., Britain, and Korea, found a clear link between child sexual abuse and suicide attempts [52,53,68,69,70]. Heins et al. [71] found that childhood trauma was correlated with psychotic disorders. Additionally, abuse was found to increase the risk for bulimia [72], substance abuse [73], obesity [46], and unintended teenage pregnancy [74].

## 6. The Psychosocial Consequences of ACE

Trauma such as ACEs can impact individuals, families, and even entire communities. Societies characterized by high poverty rates were found to have more marginalized people than communities with low poverty rates [75]. Poverty was linked to mental health difficulties in communities characterized by substance abuse, economic hardship, and violence in and out of the house [76]. Unfortunately, in such communities, people are less informed and are less aware of the factors that may improve their lives, including social connections, intimacy, and community [9,11]. Without a support system to facilitate resilience among individuals, it may be difficult for families to overcome community-related obstacles [10].

Social conditions, or the social environment one lives in, characterize the systemic barriers that create disadvantages because of the community [12]. Specifically, individuals with low socioeconomic status who experience poverty are at an increased risk for exposure to community violence and substance use, as well as social and community difficulties and obstacles [77]. The environment can positively or negatively influence education, employment, income, and healthy behaviors. The ability to navigate in one’s society is highly related to one’s ability to survive and thrive [78].

ACEs negatively affect psychological and psychosocial mechanisms, which results in cognitive and affective disturbed processing, leading to focusing on threatening stimuli [79], increasing loneliness [80], and aggressive behaviors [81]. ACEs may increase the risk of developing depression [82,83], PTSD [65], borderline personality disorder (BPD) [84], and substance abuse [85,86]. The research compared individuals with mental disturbances based on whether they were subjected to ACEs [87]. It was found that those exposed to ACEs appeared to develop earlier psychological disturbances with more severe symptomatology [88] and did not respond well to standard treatments [89] compared to those who had an unremarkable childhood. Schoedl et al. [90] found an association between the age of trauma exposure and the likelihood of developing severe depressive or PTSD symptoms in adulthood.

## 7. Long-Term Effects of ACEs

Exposure to ACEs is linked to many negative outcomes in adulthood, such as an increased risk of poverty and homelessness [91]. Moreover, a systematic review of 29 studies in the U.S., Canada, and the U.K. found a positive association between homelessness and suicide behaviors, substance misuse, and adult victimization [92]. ACEs are also commonly associated with chronic health problems in adulthood, including ischemic heart disease, liver disease, cancer, lung disease, and skeletal fractures. These ailments result from increased health risk behaviors that result from exposure to early adverse experiences [2]. Additionally, a study that looked at criminals and their ACE history while growing up found that nonsexual child abusers, domestic violence offenders, sexual offenders, and stalkers all reported four times more ACEs in comparison to a normative sample [93].

Research has indicated a significant link between exposure to trauma, subsequent stress, and poor outcomes across the lifespan affecting mental and physical health e.g., [33]. Research on ACEs has informed medicine and psychology, alerting them to the psychological underpinnings that often accompany behavior and health. While one’s childhood experiences may not be evident in adulthood, previous evidence indicates that personal history may be embedded within a spiral of emotional pain and subsequent health risk behaviors in unfavorable circumstances.

“Along with immediate health and educational effects…ACEs have been linked to higher risks of health-harming behaviors, including smoking, harmful alcohol consumption, and drug use… Exposure to ACEs is also associated with an increased risk of mental illness and other conditions, including cancer and cardiovascular disease…. The effect of ACEs on mental health and adoption of health-harming behaviors is one set of mechanisms connecting ACEs to chronic ill health” which affects neurological, hormonal, and immunological development [35], p. E518. Adverse Childhood Experiences are associated with physiological conditions related to cancer, cardiovascular disease, and respiratory disease [94]. Research has indicated that approximately one billion children aged 2–17 years are victims of violence and abuse [35]. In an example of how ACEs might perpetuate violence later in life, it was found that globally, about one-third of all women have been targeted in intimate partner violence [95]. Intimate partner violence (IPV) commonly occurs in households where children are present and may witness it and are, consequently, negatively affected by it [96,97].

Epidemiological and neurobiological studies have found that ACEs, including sexual and physical abuse, are significantly associated with brain dysfunctions, negatively affecting mental and physical health throughout one’s life [98,99,100]. It was found that the prevalence rates in Europe were 23% for physical abuse, 29% for emotional abuse, 9.6% for sexual abuse, and 18.4% for neglect [101]. Herzog and Schmahl estimated the numbers to be higher since they opined that many unreported cases exist [102]. The research found that those who underwent abuse are at a higher risk for developing mental and somatic disorders throughout their lifespan [1,91,103].

Dube et al. found that the risk of depression, suicide attempts, multiple sexual partners, contracting venereal diseases, and alcoholism increased significantly when the individual suffered ACEs [85]. The authors commented that “the remarkably consistent relationships between the ACE score and the risk of a variety of health behaviors and outcomes suggest that the mechanisms by which these stressful experiences exert their effect are resistant to, or unaffected by, the many and changing influences on health and behavior such as those that occurred throughout the 20th century.”[85], p.274. They observed that child abuse and neglect could adversely affect the developing brain, which may precipitate emotional, social, and cognitive impairments, increasing the risk for troublesome and self-harming behaviors.

## 8. Conclusions

This review aimed to provide an overview of the consequences of child maltreatment while drawing on ACEs and associated research to highlight its implications for human development. It is important for researchers, clinicians, and healthcare workers to be made aware of the complexities associated with child maltreatment and its potentially widespread effects on nearly all domains of life. While there may be various causes of ACEs, it is obvious from the reviewed literature that their consequences and effects can be destructive to the individual. For instance, unraveling these experiences may reveal a subsequent impact on mental and physical health and on one’s relationship, including intimate partner violence, sexual dysfunctions, and lack of trust and closeness. Furthermore, ACEs may also reveal society’s responsibility for in mitigating historical and intergenerational trauma by incorporating strategic implementation strategies and enhancing resources for those individuals disproportionately at a socioeconomic disadvantage. It is recommended that greater emphasis be placed on this issue to reduce the perpetuation of trauma across time and generations—i.e., if individuals can heal, it may be likely that the lives of those they encounter throughout their lifespan may be less affected, thus breaking a toxic cycle.

As this review pointed out, the implications of ACEs are profound, far-reaching, and sometimes devastating. This review is offered with the hope that clinicians, academicians, and particularly community healthcare workers be not only aware of the implications of ACEs, but also fuel their motivation to educate parents regarding avoiding ACEs in their homes and rectifying the situation if it, indeed, occurred. To conclude, this review highlighted that ACEs may have significant and even critical consequences, and that individuals and society at large must, if not prevent child abuse, at least treat its ‘scars’ and promote healing of body and soul.

## Data Availability

Not applicable.

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
