# Peer review of "The Consequences of Child Abuse"

_healthcare, 2023, doi:10.3390/healthcare11111650_

Round 1

Reviewer 1 Report

Missing method description

Lots of references, but not up to date

Author Response

Missing method description

  • That was corrected and a Method section is now included

Lots of references, but not up to date

  • Updated references now added. Most recent reference is Shappel & Tsur, 2023.

Reviewer 2 Report

1. The article does not include any new insights, it is a mere summary of existing research

2. No critical stance is taken

3. No methodology explained

4. The structure unclear and confusing. A logical structure should clearly follow the ACE pyramid since the article starts by mentioning the 8-tier pyramid and discuss the issues that are related to it

5. The article does not include a conclusion. 

Author Response

  1. The article does not include any new insights, it is a mere summary of existing research

* The present article is a thematic review of the literature, which aims to provide the reader a review of the related literature, thus providing a picture of the current state of research on the topic.

  1. No critical stance is taken

* Kindly see our response above.

  1. No methodology explained

* That was now corrected and a Methodology section was now added.

  1. The structure unclear and confusing. A logical structure should clearly follow the ACE pyramid since the article starts by mentioning the 8-tier pyramid and discuss the issues that are related to it

  • Agreed that the structure was a bit unclear. We have moved around some of the paragraphs to help the article flower better. With regards to following the ACE pyramid, we don’t believe this is necessary. The pyramid is a framework for how trauma and its effects can potentially progress across the lifespan, however, this progression is not so clear-cut or linear; thus, the aim of this review was to provide insight into potential risk factors and associated consequences.
  1. The article does not include a conclusion.

* A conclusion was now added

Reviewer 3 Report

The article provides an overview of the consequences of early adverse experiences across various domains of life, and organizes the possible directions of consideration in this problem area. The text collects and confronts conclusions from many source publications in order to systematize a certain area of research. Therefore, it is worth publishing.

I think that the introduction should be refined, in which I would expect the purpose of the text to be defined and the means of achieving it. To a greater extent, it is about presenting the methodology of developing the article.

I would also expect a separate section with conclusions. The last paragraph is very laconic and at the same time you do not need to read the whole text to be able to formulate such conclusions. I propose to indicate the implications, maybe empirical, possibly gaps or shortcomings of knowledge in relation to the discussed area, or maybe the need to verify some specific dependencies.

Author Response

The article provides an overview of the consequences of early adverse experiences across various domains of life, and organizes the possible directions of consideration in this problem area. The text collects and confronts conclusions from many source publications in order to systematize a certain area of research. Therefore, it is worth publishing.

I think that the introduction should be refined, in which I would expect the purpose of the text to be defined and the means of achieving it. To a greater extent, it is about presenting the methodology of developing the article.

I would also expect a separate section with conclusions. The last paragraph is very laconic and at the same time you do not need to read the whole text to be able to formulate such conclusions. I propose to indicate the implications, maybe empirical, possibly gaps or shortcomings of knowledge in relation to the discussed area, or maybe the need to verify some specific dependencies.

  • A conclusion section was added.

Reviewer 4 Report

Dear authors,

Please find attached the review report.

Best regards.

Author Response

The article reviews and analyses the consequences of early adverse experiences of Child

Abuse in various life domains, based on a conceptual framework. The article helps to see

the implications for health, socioemotional and psychosocial well-being, relationships, as

well as cognitive functioning.

The concept of ACE should be defined before the acronym is used.

  • A good point. That was done.

The total number of articles potentially relevant to the study is sufficient. The authors

have made an exhaustive reading of the main research on the subject. However, there are

some older articles, from 2001, 2002, 2005...

The manuscript is clear and relevant and is presented in a well-structured manner

throughout the text. Information is presented appropriately.

While it is true that there are more similar studies, as it is a more current review, it is of

interest to the scientific community.

The work is adapted to the scope of the journal.

The results of the study can provide guidance to new researchers and experts in the field,

as well as to professionals working with people who have gone through the same

experiences.

The questions that have guided this study should be clarified beforehand.

  • The aim of this review was stated and clarified. Actually, no questions guided us in developing this review.

The subject matter is relevant, but more justification should be given as to why a new

study is needed, and what is new about it.

  • As we stated above, this is a review and not a study, and as such, we did not state what new things we expect to find.

It would also be interesting to point out which authors have been the most productive on

the subject, or which countries, for example. Comment on those who are most aware of

the issue. Most cited documents and their main results.

  • Thanks for this suggestion, but we feel that it is way beyond the scope of this review.

It is not clear to me what inclusion/exclusion criteria the authors have considered in

selecting the literature on which they have relied.

  • We have explained how we searched for appropriate articles, and included the relevant ones that addressed any of the topics that were covered in this review. The selection was as comprehensive as we could make it.
  • Perhaps the concept of resilience (very important) should be explored further. Specifically

in the section where it is stated: "Without resilience, child abuse or neglect can

significantly affect well-being and quality of life and result in health risk behaviours

(Srivastav et al., 2020).

  • We added a paragraph addressing resilience, as per your suggestion.

Be careful when citing sources from a single study. It would be preferable for the authors

to look for more information to contrast. Specifically in the section they comment: "For

example, a study found that a relationship between parental antipathy and negative loving

can increase the pathology of antisocial personality disorder (Ling et al., 2022).

  • Right on, but we only gave this study as an example, and so we stated.

Section 6 on psychosocial consequences should be expanded.

  • We did that, as suggested

I found section 3 of ACEs and Psychosocial Well-being very interesting.

The article ends rather abruptly. A discussion and conclusions section is needed. It would

be interesting to be able to summarise all the information gathered.

A conclusion was added, as per your suggestion.

The fact that the authors have relied on articles published in reputable scientific journals

indicates that the research is well conducted with a sound theoretical basis. However,

there could be a bias in the selection of articles.

The results suggest that this field of research may continue to be a topic of great interest

to researchers in the coming years.

As I have said, the article could be improved with the comments I have made above.

I consider accepting the manuscript in its present form with the minor suggestions I

comment on

A fair point about bias in the selection of articles; however, this bias is unlikely to be present. The purpose of the paper is to provide a comprehensive review of ACEs and their consequences; we did not aim to compare and contrast empirical perspectives on the subject.

Round 2

Reviewer 2 Report

Line  22 : far “researching” : I guess it should be far “reaching”

Line 23: “aims” instead of “aimed”

Line 24: maltreatment is not synonymous to abuse, as mentioned by the authors it includes also neglect. 

Line 33: a framework “invented”, better a framework “created”. Please check the construction of the sentence (32-34)

Line 44: which can result neurodevelopmental : add “in”

Line 111, replace “A “by “a”

Line 113-120 : please check sentence constructions 

Line 241 : having reviewed many studies : give the right numbers

Line 250 ; add “child” sexual abuse

Line 251-252 : bullying is not only between parents and children, also not “basically” : please reframe

Line 260 : pregnancy does not fit under the title psychiatric consequences 

Line 303 : which association?

Line 317 : "between trauma"  do you mean  childhood trauma?

Point 7 on Long-term effects includes many factors already mentioned above. 

Conclusion 

It would be good to indicate in what sense exactly ACE’s is contributing to the insights in the consequences on Child Maltreatment.  ACE’s include more  experiences than child maltreatment, so can all consequences of ACE’s be attributed to the child maltreatment? Maybe the authors use the pyramid that has been mentioned above 

Author Response

Comments and Suggestions for Authors

Line  22 : far “researching” : I guess it should be far “reaching”

Line 23: “aims” instead of “aimed”

Line 24: maltreatment is not synonymous to abuse, as mentioned by the authors it includes also neglect. 

Line 33: a framework “invented”, better a framework “created”. Please check the construction of the sentence (32-34)

Line 44: which can result neurodevelopmental : add “in”

Line 111, replace “A “by “a”

Line 113-120 : please check sentence constructions 

Line 241 : having reviewed many studies : give the right numbers

Line 250 ; add “child” sexual abuse

Line 251-252 : bullying is not only between parents and children, also not “basically” : please reframe

Line 260 : pregnancy does not fit under the title psychiatric consequences 

Line 303 : which association?

  • All the above points were attended to and corrected.

Line 317 : "between trauma"  do you mean  childhood trauma?

  • We were unable to find it, not in line 317, or anywhere around it.

Point 7 on Long-term effects includes many factors already mentioned above. 

  • We are aware of it. We chose to summarize it under point 7.

Conclusion 

It would be good to indicate in what sense exactly ACE’s is contributing to the insights in the consequences on Child Maltreatment.  ACE’s include more  experiences than child maltreatment, so can all consequences of ACE’s be attributed to the child maltreatment? Maybe the authors use the pyramid that has been mentioned above 

  • Thanks for the suggestion. We have re-summarized ACEs as a conceptual framework for understanding the consequences of child maltreatment.
